# Fast Voxel-Wise Kinetic Modeling in Dynamic PET using a Physics-Informed CycleGAN

**Christian Salomonsen**
UiT The Arctic University of Norway
Tromsø, Norway
christian.salomonsen@uit.no

**Samuel Kuttner**
University Hospital of North Norway
Tromsø, Norway
samuel.kuttner@uit.no

**Michael Kampffmeyer**
UiT The Arctic University of Norway
Norwegian Computing Center
Tromsø, Norway
michael.c.kampffmeyer@uit.no

**Robert Jenssen**
UiT The Arctic University of Norway
Norwegian Computing Center & Univ. of Copenhagen
Tromsø, Norway
robert.jenssen@uit.no

**Kristoffer Wickstrøm**
UiT The Arctic University of Norway
Tromsø, Norway
kristoffer.k.wickstrom@uit.no

**Jong Chul Ye**
Korea Advanced Institute of Science and Technology
Daejeon, Republic of Korea
jong.ye@kaist.ac.kr

**Elisabeth Wetzer**
UiT The Arctic University of Norway
Tromsø, Norway

## Abstract

Tracer kinetic modeling serves a vital role in diagnosis, treatment planning, tracer development and oncology, but burdens practitioners with complex and invasive arterial input function estimation (AIF). We adopt a physics-informed CycleGAN showing promise in DCE-MRI quantification to dynamic PET quantification. Our experiments demonstrate sound AIF predictions and parameter maps closely resembling the reference.

## 1 Introduction

Kinetic modeling in dynamic positron emission tomography (dPET) allows for the determination of physiological parameters that are not accessible in static PET imaging. The parameters derived from kinetic modeling describe the underlying metabolic processes in the body, and can be used for diagnosis and treatment planning in various diseases, including cancer, neurological disorders, and cardiovascular diseases [6, 18, 3]. For instance, in oncology, kinetic modeling can help differentiate between benign and malignant lesions, assess tumor aggressiveness, and monitor treatment response in theranostics, providing superior information compared to static PET imaging.

To derive these parameters, a tracer kinetic model is fitted to the time-activity curves (TACs) obtained from the dPET images. This fitting process requires an accurate arterial input function (AIF), which represents the concentration of the tracer in the blood plasma over time [1]. The gold-standard method for AIF determination is through arterial blood sampling, which is invasive, labor-intensive, and infeasible for routine clinical use. While alternatives such as image-derived input functions (IDIFs) [12, 5, 7, 4] and population-based input functions (PBIFs) [17], and more recently, machine

learning [10, 11], and deep learning methods [9, 14] offer noninvasive alternatives, these methods often suffer from inaccuracies due to partial volume effects, motion artifacts, and inter-subject variability. Moreover, once the AIF is determined, solving the kinetic model is typically performed over predetermined regions of interest, instead of on a voxel-level, due to the computationally intensive nature of nonlinear fitting algorithms. While voxel-wise kinetic modeling is possible, and used in brain quantification [2], it is not commonly used in whole-body PET imaging due to the computational burden compounded by the low signal-to-noise ratio (SNR) in many tissues.

Due to the AIF being a vital component in parametric imaging, yet only useful for this purpose, simultaneous estimation of both quantities with an efficient feedforward neural network during the inference phase presents a more streamlined approach. Furthermore, a physically-informed approach that leverages the underlying physics captured in the kinetic models have shown promise in out-of-distribution settings [15]. Thus, we adopt a physics-informed CycleGAN as recently proposed for AIF estimation in the context of dynamic contrast-enhanced magnetic resonance imaging (DCE-MRI) [13]. This approach uses unpaired images and kinetic parameter maps with a combination of an adversarial loss and a cycle consistency loss, promoting consistency in estimated parameter maps with respect to the parameters of the input imaging data. This approach enables direct learning of the mapping from PET time series to kinetic parameter maps, eliminating the need for AIF determination and expensive fitting procedures.

## 2 Methods

Our approach closely follows that of Oh et al. [13], which uses a CycleGAN to learn the mapping between DCE-MRI and their parametric images, exploiting domain knowledge of the underlying kinetic model describing their relationship, where we use dPET data and a PET-specific compartment model instead. As the original CycleGAN [19], the proposed approach does not require paired data. The CycleGAN consists of a generator $G : X \to Y, C_A$ that maps the dPET images $X$ to the kinetic parameter maps $Y$ and AIF $a$, and a forward tracer kinetic model $F : Y, C_A \to X$ that maps the kinetic parameter maps back to the dPET images. The forward model uses an irreversible two-tissue compartment model (2TCM) [16] to generate the dPET images from the kinetic parameter maps and AIF. The model is defined as follows:

$$X = C_{PET}(t) = V_b \cdot C_A(t) + (1 - V_b) \cdot C_T(t)$$
$$C_T(t) = \frac{K_1}{k_2 + k_3}[k_3 + k_2 \cdot e^{-(k_2+k_3)t}] \otimes C_A(t) \tag{1}$$

Where $C_{PET}(t)$ is the tissue TAC observable from the scan, $C_A(t)$ is the AIF, $C_T(t)$ denotes the tissue compartment concentration, $V_b$ is the blood volume fraction. Rate constants $K_1$, $k_2$, and $k_3$ represent the tracer transport from blood to tissue, from tissue to blood, and the phosphorylation rate, respectively. The symbol $\otimes$ denotes the convolution operation. For further details, we refer the interested reader to Oh et al. [13].

### 2.1 Data

This study used a dataset of 70 whole-body [$^{18}$F]FDG dPET scans of mice accompanied by arterial blood sampling simultaneously acquired during scanning. After reconstruction, the spatio-temporal images were of dimensions $42 \times 96 \times 48 \times 48$ (time, axial, coronal, sagittal) with a time frame duration of $1 \times 30$s, $24 \times 5$s, $9 \times 20$s, and $8 \times 300$s. To obtain the initial kinetic maps for each scan, a voxel-wise kinetic model using a linearization of the 2TCM [2] was fitted to the dPET images using the arterial input function obtained from blood sampling, solving Eq 1 to determine the rate constants.

### 2.2 Implementation Details

Unpaired training is accomplished by sampling a dPET image and a parametric map with AIF from two distinct samples. The patch discriminator is initialized with 32 filters in the final layer, and 3 layers in total. The generator takes 3D PET images with 42 input channels, to match the time dimension of our data, before mapping to 4 output channels $K_1, k_2, V_b, k_3$ corresponding to the output produced by the forward compartment model (Eq. 1). The CycleGAN is trained over 1000 epochs using AdamW [8] with $\beta_1 = 0.5$ and $\beta_2 = 0.999$.

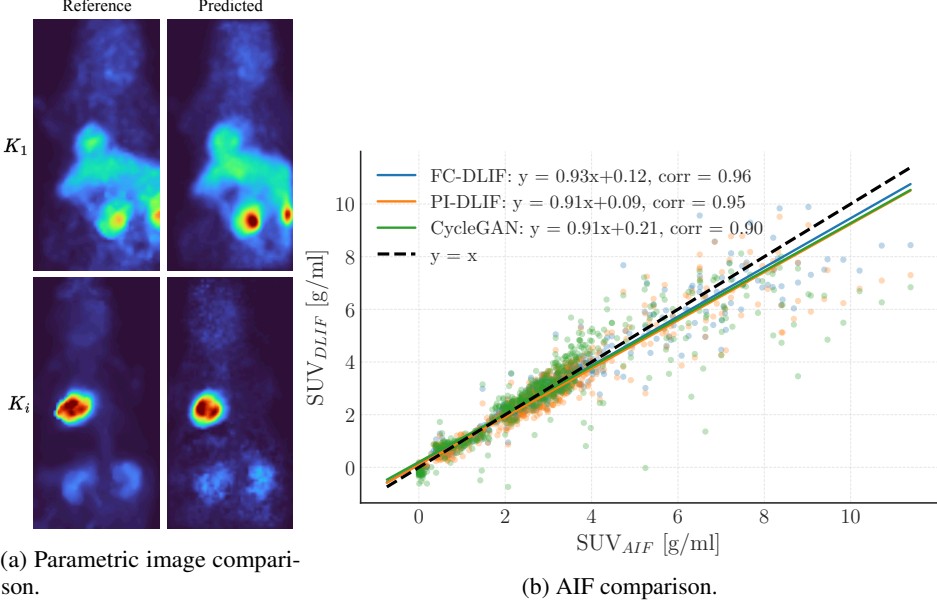

(a) Parametric image comparison.

(b) AIF comparison.

Figure 1: Comparison of (a) the ground truth and estimated kinetic parameter maps for $K_1$ and $K_i$, and (b) the estimated AIF from FC-DLIF [14], PI-DLIF [15] and the proposed CycleGAN approach. In (a), the first axis contains the measured AIF values from blood sampling, while the second axis contains the estimated AIF values from the three methods. The dashed line reports the identity line, i.e., perfect estimation.

## 3  Experiments and Results

We evaluate fidelity to our reference parametric mapping pipeline on a held-out test set using structural similarity index measure (SSIM) and peak signal-to-noise ratio (PSNR), obtaining $0.7425$ and $34.49$ dB, respectively. Because the network is trained to produce the reference parametric maps, these metrics reflect generalization to the baseline rather than absolute physiological accuracy. The perceived quality of the produced parametric maps (Fig. 1a) reflects the decent SSIM and PSNR values reported, but notable discrepancies in intensity and fine details are observed. Our implementation continue using instance normalization in the generator [13], which interferes with the absolute scale of the rate constants because of the per-volume normalization being applied.

In terms of AIF estimation, the CycleGAN is compared against two recently proposed methods for AIF estimation from dPET images: a fully convolutional deep learning-based input function (FC-DLIF, [14]) predictor, and a physics-informed extension to this model (PI-DLIF, [15]). These methods have undergone more extensive training and predict using an ensemble of 10 models, yet the CycleGAN achieves comparable performance, as shown in Fig. 1b.

## 4  Discussion and Conclusion

In this work, we have presented a physics-informed CycleGAN for simultaneous estimation of the arterial input function and kinetic parameter maps from [$^{18}$F]FDG dPET images. The method learns to map dPET images to kinetic parameter maps and AIFs without requiring paired training data. The method achieves comparable AIF estimation performance to two recently proposed deep learning-based methods, and exhibit a high level of fidelity to the reference parametric mapping pipeline. However, normalization in the generator interferes with the absolute scale of the estimated rate constants, which is a limitation of the current implementation. Future work will focus on addressing this shortcoming, artificially augmenting the scant data available, and evaluating the method on out-of-distribution data.

# 5 Potential Negative Societal Impact

The proposed methodology has been tested in pre-clinical studies, on a limited set of healthy mice. More research is needed until the proposed approach can eventually be translated to clinical use-cases in human studies and cancer diagnostics. To address potential negative impacts associated with this application development of XAI and uncertainty estimates are necessary for spatio-temporal images that can capture underlying dynamics in order to identify outliers and failure cases. As the proposed method is a generative method with a reconstruction component, care has to be taken regarding weight sharing, as the weights encode potentially sensitive medical data if applied to human patients.

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
