# OpenReview forum: "Fast Voxel-Wise Kinetic Modeling in Dynamic PET using a Physics-Informed CycleGAN"
_EurIPS.cc/2025/Workshop/MedEurIPS — EurIPS 2025 Workshop MedEurIPS Submission_

### Official Review · Reviewer_Phff · 2025-10-29
**A well-motivated and methodologically clear study with some missing evaluation.**

**Rating:** 7
**Confidence:** 4

**Review:**

Strengths:
-	Relevant and well-motivated approach.
-	The voxel-wise approach allows for a high localized resolution
-	Promising performance compared to baseline methods.
-	 Clear methodological discussion and evaluation
Weaknesses:
-	The title claims a “fast” method, but no runtime comparison was done.
-	In line 72, the authors argue that they don’t report a measure for the “absolute physiological accuracy”. I think it would be nice to add also a measure for this, like MSE.

---

### Official Review · Reviewer_8PS6 · 2025-11-03
**The paper is using an recently proposed physics-informed CycleGAN for arterial input function (AIF) estimation in the context of dynamic contrast-enhanced magnetic resonance imaging and apply it to dynamic positron emission tomography (dPET). The approach is trained and evaluated on mice data and compared to other methods for AIF estimation from dPET. The paper has a novel application, but could be clearer in terms of methods and results.**

**Rating:** 7
**Confidence:** 4

**Review:**

**Positive Aspects**
- The paper addresses a novel and relevant application in the field of PET imaging.
- It tackles an important and clinically meaningful problem.
- The authors present strong results with convincing comparisons against state-of-the-art methods.

**Major Comments**
- Voxel-wise Modeling Justification:
The authors state: “While voxel-wise kinetic modeling is possible […] it is not commonly used in whole-body PET imaging due to the computational burden compounded by the low signal-to-noise ratio (SNR) in many tissues.”
However, the proposed method also adopts a voxel-wise approach. This apparent contradiction should be addressed. Please discuss the feasibility of your voxel-wise strategy in the context of SNR and computational load, and explain how the method overcomes the challenges outlined.
- Clarification of Method Adaptations from [13]:
The paper mentions: “we adopt a […]” with reference to method [13].
However, it remains unclear how the proposed approach differs from or extends the original method in [13]. Please briefly summarize the modifications in the introduction.
In the methods section, provide a more detailed explanation of the changes, especially regarding the handling of the arterial input function (AIF). For instance, [13] does not incorporate AIF, while the proposed method appears to use it (as suggested by the unpaired training description: “sampling a dPET image and a parametric map with AIF from two distinct samples”). How exactly is AIF integrated into the model, and what implications does this have for performance and applicability?
- AIF Estimation and Stated Limitations:
The paper proposes a “simultaneous estimation of the arterial input function and kinetic parameter maps from [18F]FDG dPET images,” yet also highlights that AIF alternatives “often suffer from inaccuracies due to partial volume effects, motion artifacts, and inter-subject variability.”
This tension should be clarified: What advantages does the proposed approach offer over traditional AIF estimation methods, and how robust is it to the limitations mentioned?

**Minor Comments**
- The abbreviation DCE-MRI is used in the abstract but not defined. Please introduce it upon first use.
- The sentence “comparable performance, as shown in Fig. 1b.” is vague. Please elaborate briefly on the comparison and key findings shown in the figure to make the point more informative.

---

### Decision · Program_Chairs · 2025-11-03

**Decision:**

Accept (Poster)

**Comment:**

Both reviewers highlight that the paper tackles a clinically meaningful and novel application of physics-informed CycleGANs to dynamic PET and find the results convincing.